# Tetrathiomolybdate induces dimerization of the metal-binding domain of ATPase and inhibits platination of the protein

Tiantian Fang [1], Wanbiao Chen[2], Yaping Sheng[1], Siming Yuan[1], Qiaowei Tang[3], Gongyu Li[1], Guangming Huang [1], Jihu Su[3], Xuan Zhang[2], Jianye Zang[2] & Yangzhong Liu [1]

Tetrathiomolybdate (TM) is used in the clinic for the treatment of Wilson's disease by targeting the cellular copper efflux protein ATP7B (WLN). Interestingly, both TM and WLN are associated with the efficacy of cisplatin, a widely used anticancer drug. Herein, we show that TM induces dimerization of the metal-binding domain of ATP7B (WLN4) through a unique sulfur-bridged $Mo_2S_6O_2$ cluster. TM expels copper ions from Cu-WLN4 and forms a copper-free dimer. The binding of Mo to cysteine residues of WLN4 inhibits platination of the protein. Reaction with multi-domain proteins indicates that TM can also connect two domains in the same molecule, forming Mo-bridged intramolecular crosslinks. These results provide structural and chemical insight into the mechanism of action of TM against ATPase, and reveal the molecular mechanism by which TM attenuates the cisplatin resistance mediated by copper efflux proteins.

[1] CAS Key Laboratory of Soft Matter Chemistry, Department of Chemistry, University of Science and Technology of China, Hefei, Anhui 230026, China. [2] Hefei National Laboratory for Physical Sciences at Microscale CAS Center for Excellence in Biomacromolecules, Collaborative Innovation Center of Chemistry for Life Sciences, and School of Life Sciences, University of Science and Technology of China, 96 Jinzhai Road, Hefei, Anhui 230026, China. [3] Department of Modern Physics, University of Science and Technology of China, Hefei, Anhui 230026, China. These authors contributed equally: Tiantian Fang, Wanbiao Chen.  Correspondence and requests for materials should be addressed to X.Z. (email: xuanzbin@ustc.edu.cn) or to J.Z. (email: zangjy@ustc.edu.cn) or to Y.L. (email: liuyz@ustc.edu.cn)

P-type ATPases, including ATP7A and ATP7B, are responsible for active copper efflux that is essential for maintaining copper homeostasis in mammalian cells[1]. These two proteins are also known as Menkes (MNK) and Wilson (WLN) disease proteins, since dysfunction of ATP7A and ATP7B leads to these diseases, respectively. Interestingly, these Cu-ATPases are also associated with resistance to cisplatin, one of the most widely used anticancer drugs in the clinic[2,3]. Overexpression of Cu-ATPases is a feature of cisplatin-resistant cancer cells[3–6], while silencing of Cu-ATPase genes recovers drug sensitivity[7]. Mechanistic studies revealed that Cu-ATPases are involved in the efflux/sequestration of cisplatin[2,6]. ATPases contain six metal-binding domains (MBDs) that share high sequence homology with Atox1, a chaperone delivering copper to ATPase, and exhibit similar copper-binding properties[8]. Cisplatin can directly bind to the MBDs of ATPases at the Cu-binding site, and MBDs are required to modulate resistance to cisplatin[9,10]. These findings suggest that the inhibition of cisplatin binding to the MBDs of ATPases could be an effective approach to circumvent drug resistance of cisplatin.

Ammonium tetrathiomolybdate ($[(NH_4)_2MoS_4]$, TM) is a copper chelator used in the clinic for the treatment of Wilson's disease, a copper metabolism disorder disease associated with the WLN protein[11]. Co-administration of TM improves the efficacy of cisplatin, and reduces drug resistance[12]. The molecular mechanism of this synergistic effect is unknown. Crystal structure analysis revealed that TM binds to Atx1, an analog of Atox1 in yeast, and forms a stable trimeric protein complex [TM·Cu·(Cu-Atx1)$_3$][13]. On the other hand, our recent research showed that TM can inhibit platination of the copper chaperone Atox1[14]. Since the MBDs of ATPases exhibit similar protein folding and reactivity to Atox1 (such as copper-binding and cisplatin reactions)[8,15], it could be anticipated that TM might also induce the formation of Mo-Cu-centered trimeric protein complexes of MBDs, and therefore inhibit the reaction of cisplatin.

In this work, we investigate the reaction of TM with MBDs of ATPase, and effect of TM reaction on the platination of the proteins. We show that the reaction of TM with WLN4 (the fourth MBD of Wilson's disease protein) generates a protein dimer, and TM binding results in copper ejection from the Cu-WLN4 protein. Unlike the Cu-Mo core formed in the reaction of Atx1, a copper-free $Mo_2S_6O_2$ center is generated in the reaction of WLN4. The two copper-binding residues, Cys14 and Cys17, which are also the binding site of cisplatin, directly coordinate Mo and are involved in the formation of the $Mo_2S_6O_2$ core in this protein dimer. Further studies confirm that the formation of this protein dimer inhibits the reaction of cisplatin, which explains how TM reduces ATPase-related cisplatin resistance.

## Results

**TM induces dimerization of the MBDs of ATPases.** The reaction of TM with Cu-WLN4 generated a yellow complex, and the product was purified by size-exclusion chromatography (SEC) (Supplementary Fig. 3a). Pale yellow crystals were obtained using the sitting-drop vapor-diffusion method (Supplementary Fig. 3b). The crystal structure of the TM-WLN4 complex was determined at 1.6 Å resolution (Supplementary Table 1). The complex forms a protein dimer linked by a $Mo_2S_6O_2$ core (Fig. 1a). Two Mo atoms are bridged by two sulfur atoms, and each protein subunit provides two thiol groups for the Mo-coordination. The Mo atom occupies the copper-binding residues (Cys14 and Cys17), preventing copper ions coordinating to this site. Inductively coupled plasma mass spectrometry (ICP-MS) measurements showed that no copper was present in the crystal of Mo-WLN4 (Supplementary Fig. 3c). This result indicates that Mo binding leads to

copper release from the protein, in contrast with the reaction of Atx1 with TM, which forms a TM-Cu-Atx1 ternary complex[13].

The detailed coordination structure of the TM-WLN4 complex is shown in Fig. 1b. The Mo atom is coordinated via a pyramidal geometry, and all eight distances between Mo and S atoms (2.3 −2.5 Å) are in the typical range for Mo–S covalent bonds. The two distances between Mo and O (1.5 and 1.6 Å) indicate the formation of Mo = O double bonds.

Since the TM-induced dimerization of Cu-WLN4 in the crystal is rather different from the reaction of Cu-Atx1, we performed size-exclusion chromatography coupled with multi-angle light scattering (SEC-MALS) to further characterize the overall solution status of the protein complex. Both apo- and Cu-WLN4 are present mainly as a monomer in solution, since the peaks appeared with the retentions corresponding to a molecular weight of 7.4 and 7.5 kDa for apo- and Cu-WLN4, respectively (Fig. 2a, b). Incubation of Cu-WLN4 with TM resulted in a product with a molecular weight of 15.5 kDa, corresponding to a protein dimer (Fig. 2c). The narrow peak in the SEC profile indicates a highly homogeneous distribution of the protein complex in solution. This result confirmed that the protein dimer observed in the crystal structure is also present in solution.

Since copper was not observed in the crystal structure of the TM-WLN4 complex, this indicates that the reaction with TM expelled copper ions from Cu-WLN4, suggesting the dimerization of WLN4 does not require copper ions. To verify this assumption, we reacted apo-WLN4 with TM, and tested whether TM can react with apo-WLN4 to generate the same protein dimer. SEC-MALS analysis showed that incubation of apo-WLN4 with TM generated a product with a molecular weight of 15 kDa, indicating that reaction of apo-WLN4 also results in a protein dimer (Fig. 2d). In addition, we also conducted nuclear magnetic resonance (NMR) relaxation measurements on apo-WLN4 and TM-WLN4. Based on the $T_1$ and $T_2$ values, the rotational correlation time ($\tau_c$) was 4.62 ns for apo-WLN4 and 10.44 ns for TM-WLN4 (Table 1), corresponding to proteins with a molecular weight of 7.7 and 17.4 kDa, respectively[16]. This result is consistent with the dimerization of WLN4 upon TM binding.

MALDI-TOF mass spectra were recorded to directly measure the change in molecular weight upon TM binding. The result confirmed the formation of the dimeric TM-WLN4 product in the reaction of apo-WLN4 (m/z: cal. 15,607 Da, obs. 15,588 Da, see Supplementary Fig. 4a and Supplementary Table 2). The TM-induced protein dimer was also detected following the reaction of other apo-MBDs, such as TM-WLN6 (m/z: cal. 16,178 Da, obs. 16,134 Da, see Supplementary Fig. 4b). Interestingly, hetero-dimers were observed following the incubation of TM with mixtures of two different apo-MBDs. For instance, WLN4-TM-WLN6 and WLN4-TM-WLN5 were formed in the reaction of apo-WLN4 with apo-WLN6 or apo-WLN5, respectively. (Supplementary Fig. 4c, d). This observation suggests that TM-induced dimerization occurs on different MBDs of ATPases, and confirms that the formation of protein dimers does not require copper ions.

To further confirm the assumption that copper ions are not needed for the TM-mediated dimerization of WLN4, we incubated apo-WLN4 with TM and solved the crystal structure of the TM-WLN4 complex (Supplementary Table 1 and Supplementary Fig. 5). Structural comparison showed that the overall dimeric structure of TM-WLN4 obtained from apo-WLN4 is identical to that from Cu-WLN4 (Supplementary Fig. 5a). The $Fo - Fc$ maps of the $Mo_2S_2O_2$ center also exhibit the same density features in both structures (Supplementary Fig. 5b, c). These results further indicate that copper ions are not required for TM-mediated dimerization of WLN4.

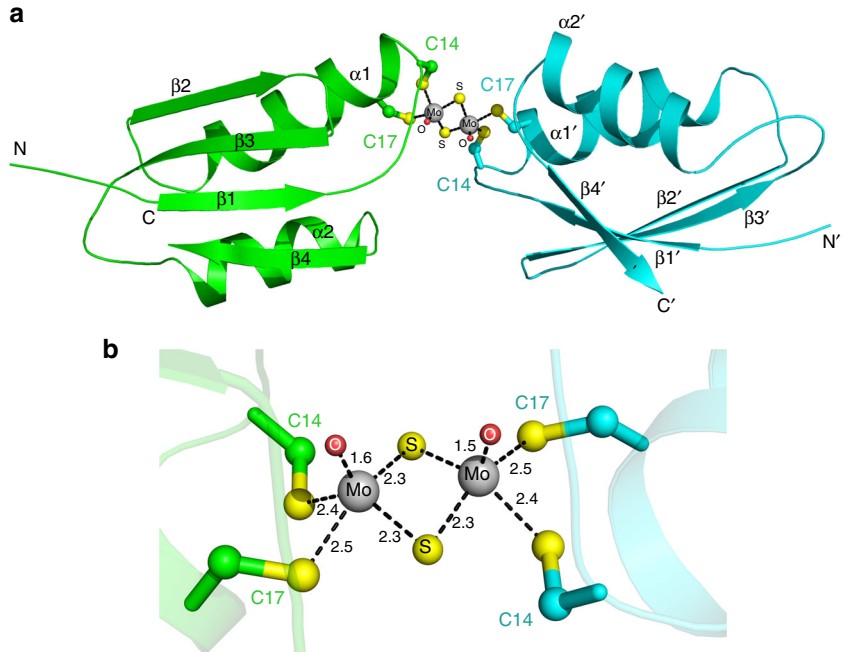

**Fig. 1** TM-mediated dimerization of Cu-WLN4. **a** Overall structure of TM-WLN4 dimer. The two subunits are colored in green and cyan, respectively. The secondary elements are labeled in black. Two Mo atoms are shown as gray spheres. S and O atoms that coordinate to Mo are colored in yellow and red, respectively. Cysteine residues that interact with Mo are shown as sticks. **b** The detailed coordination of the $Mo_2S_6O_2$ core. The coordination bonds shown as black dash and the bond lengths are labeled

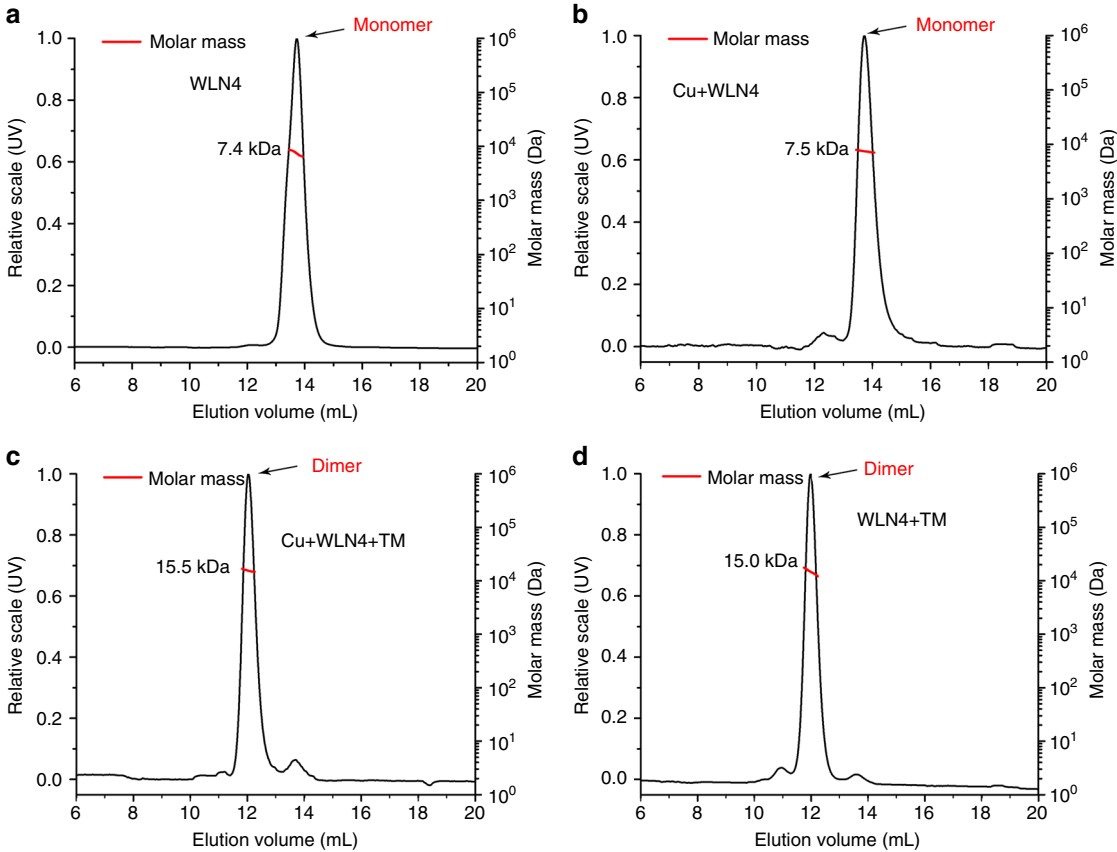

**Fig. 2** SEC-MALS results of WLN4 protein. **a** apo-WLN4; **b** Cu-WLN4; **c** TM-WLN4 generated by incubation of TM with Cu-WLN4; **d** TM-WLN4 generated by incubation of TM with apo-WLN4. The Cu-WLN4 was produced by the reaction of apo-WLN4 with [Cu(CH$_3$CN)$_4$]ClO$_4$ in 50 mM MES, 150 mM NaCl, pH 6.0. The calculated molecular weight and oligomeric state of each sample is labeled black and red, respectively

**Table 1 Estimation of molecular size of the TM-WLN4 complex according to NMR relaxation**

|  | $T_1$ (ms) | $T_2$ (ms) | $\tau_c$ (ns) | MW (kDa) |
|---|---|---|---|---|
| WLN4 | 453 ± 40 | 142 ± 19 | 4.62 | 7.7 |
| TM-WLN4 | 830 ± 124 | 72 ± 20 | 10.44 | 17.4 |

$T_1$ longitudinal relaxation time, $T_2$ transverse relaxation time, $\tau_c$ rotational correlation time

**Table 2 Copper released from Cu-WLN4 in the reaction with TM**

| Cu-WLN4:TM | 1:0 | 1:0.5 | 1:1 | 1:3 |
|---|---|---|---|---|
| Cu released (μM)[a] | 0.02 | 2.1 | 4.5 | 9.1 |

[a]Ten micromolar Cu-WLN4 was incubated with TM at 37 °C for 3 h

**TM expels copper ions from MBDs**. The absence of copper ions in the TM-WLN4 crystal structure from Cu-WLN4, together with the TM-induced dimerization of apo-WLN4, indicates that TM binding expels copper ions from the Cu-WLN4 protein. To confirm this assumption, copper release from Cu-WLN4 was measured upon the reaction with TM. After incubation of Cu-WLN4 (10 μM) with different amounts of TM, samples were concentrated by ultrafiltration (molecular weight cutoff = 3 kDa), and the copper concentration in the flow-through was measured using ICP-MS. The results showed that the reaction led to copper release in a TM concentration-dependent manner; 91% of copper was present in the flow-through upon the reaction with a threefold molar ratio of TM (Table 2). This confirmed that the binding of TM to Cu-WLN4 results in copper ejection from the protein.

The ICP-MS data in Table 2 show that an excessive amount of TM is needed to expel copper from Cu-WLN4 (equimolar TM resulted in only 45% copper release), suggesting that excess TM might chelate with the copper ions released from WLN4. To verify this assumption, we recorded ultraviolet (UV) spectra of the proteins after TM binding. In comparison with the reaction of apo-WLN4, reaction of Cu-WLN4 with TM resulted in additional shoulder peaks at ~400 nm and 520 nm (Supplementary Fig. 6). This difference is consistent with the reaction of TM with free Cu ions, which also generates a shoulder peak at this region that is absent in the spectrum of free TM. This result indicates that TM can indeed chelate the copper ions released from the protein, which could promote the binding of TM to the Cu-WLN4 protein.

**TM inhibits platination of MBDs**. The TM-induced copper ejection from WLN4 strongly suggests that TM can perturb the function of ATPases. In addition, a number of researches have shown that ATPases are involved in the cisplatin resistance, while TM can reduce the drug resistance[12,17]. Therefore, we tested whether TM binding could inhibit the platination of ATPases, which could be associated with the synergistic effect of cisplatin and TM in cancer chemotherapy. The effect of TM on the platination of WLN4 was analyzed using $^1H$-$^{15}N$ HSQC NMR spectroscopy recorded on $^{15}N$ isotope-labeled proteins. Adding TM to apo-WLN4 shifted only a few peaks in the spectra (Supplementary Fig. 7a), consistent with the crystal structure showing that TM binding does not alter the overall structure of WLN4. The reaction of apo-WLN4 with cisplatin can be clearly observed in NMR spectra, in which a large number of peaks of apo-WLN4 are shifted after incubation with cisplatin (Fig. 3a). Signal changes

were more pronounced with a longer reaction time (Supplementary Fig. 7b). Unlike the case of apo-WLN4, a nearly identical spectrum was obtained after incubation of TM-WLN4 with cisplatin (Fig. 3b). These results indicate that the binding of TM to apo-WLN4 inhibits platination of the protein.

Platination of Cu-MBD was analyzed using ESI-MS spectrometry. The platinated WLN4 was detected upon incubation of Cu-WLN4 with cisplatin, mainly in the form of [Pt(NH$_3$)$_2$·(WLN4)] (Fig. 3, Supplementary Fig. 8, and Supplementary Table 3). Interestingly, [Pt(NH$_3$)$_2$Cl·Cu(WLN4)] was detected in ESI-MS spectra ($m/z$: obs. 1608.97, cal. 1608.64), showing that copper ions remain bound to the protein while platinum binding to WLN4 (Fig. 3e). Co-existence of copper and platinum was also observed in the reaction of Atox1[17]. By contrast, no platination adducts were detected in the reaction of cisplatin with the TM/Cu-WLN4 complex (Fig. 3d). This result confirmed that formation of the TM complex inhibits the reaction of Cu-WLN4 with cisplatin.

It has been reported that platination of proteins can perturb protein folding and induce protein aggregation[18,19]. Circular dichroism (CD) spectroscopy showed that both apo- and Cu-WLN4 possess a well-folded structure, demonstrated by the intense negative band at 220 nm (Fig. 4a, b). No detectable change was observed in the CD spectrum following TM reaction, since TM-induced dimerization of WLN4 does not alter the protein secondary structure. Binding of cisplatin to WLN4 significantly reduced the negative signal at 220 nm, indicating that platination of WLN4 markedly disrupts the protein secondary structure. By contrast, cisplatin did not alter the CD spectrum of the TM-WLN4 complex. This observation is consistent with the NMR results, showing that TM binding significantly reduced the platination of WLN4 in both apo- and Cu-bound forms.

Platination-induced unfolding of MBDs can lead to protein aggregation. Thus, the products of the reactions of cisplatin with WLN4, both in the absence or presence of TM, were analyzed. Reaction with MNK3 was also investigated since both WLN and MNK proteins are associated with drug resistance to cisplatin. Gel electrophoresis showed that both apo-MBDs and Cu-MBDs appeared as monomer bands, and TM-induced protein dimerization cannot be observed on this denaturing gels. In the absence of TM, cisplatin can react with MBDs in both apo- and holoforms, leading to the oligomerization of MBDs (Fig. 4). This result is consistent with previous studies showing that the platination of Atox1 and other MBDs causes protein aggregation and polymerization[9,20]. TM binding significantly reduced the aggregation of apo-MBD and Cu-MBD; only a small amount of protein dimer was observed in the same conditions. This result confirmed that TM can prevent the platination of MBDs in both apo- and holo-forms.

**TM induces intramolecular crosslinks between MBDs**. The formation of TM-linked hetero-dimers of MBDs, such as WLN4-TM-WLN5, implies that TM could crosslink two adjacent MBDs within the same molecule (WLN contains six MBDs). In order to verify this hypothesis, we constructed multi-domain WLN constructs, including domains 1−4 (WLN1234) and domains 4−6 (WLN456). SEC analyses showed that reaction with TM generated both oligomeric and monomeric adducts of WLN1234 (Supplementary Fig. 9a). While the oligomeric species were generated due to the intermolecular crosslinking, the monomeric species could include Mo-mediated intramolecular crosslinking. Therefore, the fraction of monomer adduct was collected for further analysis. SEC-MALS analysis confirmed the monomeric composition of this fraction, possessing a molecular weight of ~39.8 kDa (cal. 40.4 kDa) (Fig. 5a). The Mo-coordination was

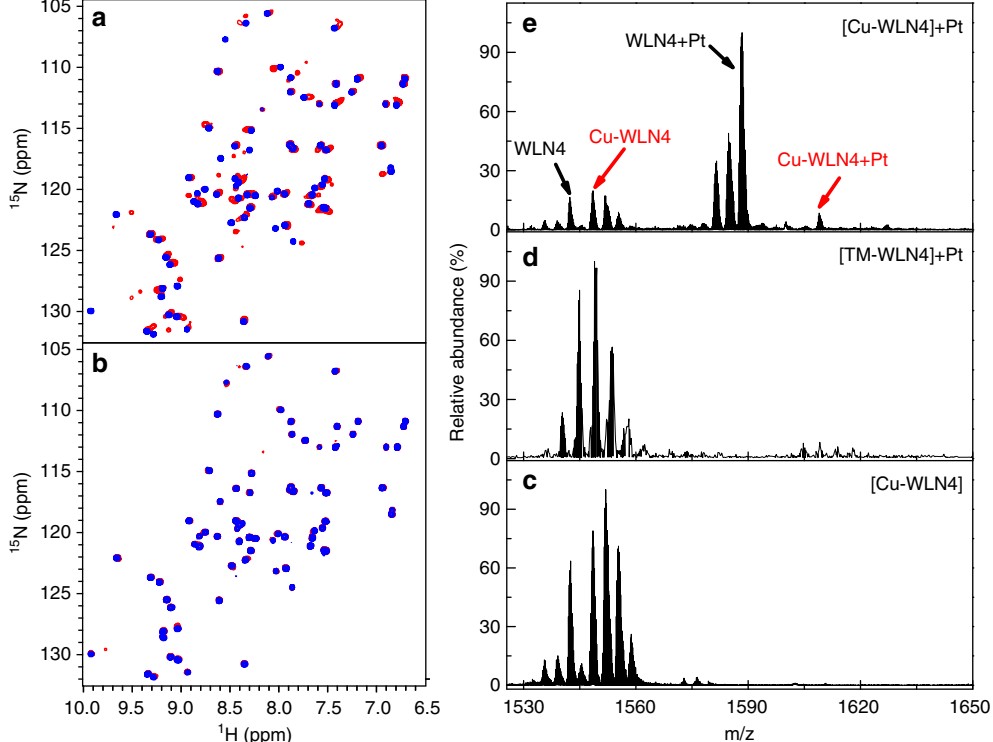

**Fig. 3** The effect of TM on platination of WLN4. **a**, **b** $^1$H-$^{15}$N HSQC NMR spectra. **a** Superposition of spectra of apo-WLN4 before (blue) and after (red) the reaction of cisplatin. **b** Superposition of spectra of TM-WLN4 before (blue) and after (red) reaction of cisplatin. **c–e** ESI-MS spectra. **c** Cu-WLN4; **d** incubation of TM/Cu-WLN4 adducts with cisplatin; **e** incubation of Cu-WLN4 with cisplatin. The reaction was conducted on 50 μM protein with 150 μM cisplatin at 25 °C for 8 h. The selected $m/z$ region shows + 5 charged peaks. The peaks are shown in a cluster due to the binding of different number of $NH_4^+$ ions since NH$_4$OAc was used. The assignment of peaks is given in Supplementary Fig. 8 and Supplementary Table 3

confirmed by UV–Vis spectroscopic measurements, in which the absorption at ~390 nm and ~320 nm was consistent with the TM-WLN4 adduct (Fig. 5b and Supplementary Fig. 9d). Moreover, ESI-MS spectra show that reaction with TM led to a mass increase of 287.4 Da on WLN1234 (from 41538.9 to 41826.3 Da), consistent with the addition of $Mo_2S_2O_2$ (cal. 288.0 Da) to the protein (Fig. 5c). These results confirmed the formation of an intramolecular $Mo_2S_6O_2$ cluster between two MBDs in the WLN1234 protein.

The reaction of TM was also conducted on WLN456 (Supplementary Fig. 9b). The SEC-MALS and UV–Vis results also revealed a monomeric species with TM binding (Supplementary Fig. 9c, f). In addition, we performed site-directed mutagenesis of various MBDs in order to locate the two Mo-crosslinked domains in the multi-domain proteins (Supplementary Figs. 1 and 2), since Cys-to-Ser mutation abolishes the Mo-coordination to MBDs of ATPases. By comparing UV–Vis spectra of different variants, we concluded that the $Mo_2S_2O_2$ cluster links domains 3 and 4 in WLN1234, because the mutation of domains 1 and 2 did not influence Mo binding, whereas Mo absorption was not observed if either domain 3 or 4 was mutated (Supplementary Fig. 9e). Mutagenesis also indicted that Mo-induced crosslinking occurs between domains 5 and 6 in WLN456 (Supplementary Fig. 9f).

## Discussion

It has been well-characterized that each MBD of ATPases contains a copper-binding site (CXXC motif). Cu-binding triggers relocalization of ATPases from the Golgi apparatus to the cell membrane, which eliminates excess copper from cells[21]. Recent researches indicate that cisplatin also binds to the MBDs of ATPases at the copper coordination site, and co-localization of copper and platinum occurs[10]. Therefore, ATPases sequester both

Cu and Pt, resulting in cross-resistance of copper and platinum in cancer cells[2,22,23]. In the present work, crystal structures of the TM-WLN4 complexes revealed that Mo atoms directly bind to WLN4 at the copper-binding site (Cys14 and Cys17). The occupation of Mo at these residues impedes the binding of copper and platinum. Hence, TM can inhibit the ATPase-mediated sequestration of cisplatin. This result elucidates the mechanism that TM reduces cellular resistance to cisplatin, and explains the synergistic effect of TM and cisplatin in cancer treatment.

The crystal structure of the TM-Atx1 adduct revealed that TM inhibits copper trafficking through the formation of the [TM·Cu·(Cu-Atx1)$_3$] complex, a trimeric protein complex containing a Mo-Cu-centered cluster. However, the reaction of WLN4 with TM generates a dimeric adduct instead of a protein trimer. In addition, binding of TM expels copper from WLN4, in contrast to the formation of the Mo-Cu core in the reaction of Atx1. These discrepancies could be due to either the different property of the two proteins, or caused by the different conditions in the TM reactions since the Mo-Cu-centered trimeric structure of Atx1 was obtained under aerobic conditions. To explore the reasons for the different structures of Atox1 and WLN4, we reacted TM with two proteins under identical conditions (containing 4.5 mM reducing agents) in this work. Interestingly, SEC-MALS analysis indicates that both apo-Atox1 and Cu-Atox1 generated dimeric adducts following reaction with TM (Supplementary Fig. 10). This result suggests that, under reducing conditions, TM can induce the dimerization of both WLN4 and Atox1, and copper is not required for the reaction. It is worth noting that the intracellular environment represents reducing conditions and contains millimolar concentrations of reducing agents; therefore, the copper-free dimeric adducts could be more relevant to physiological conditions.

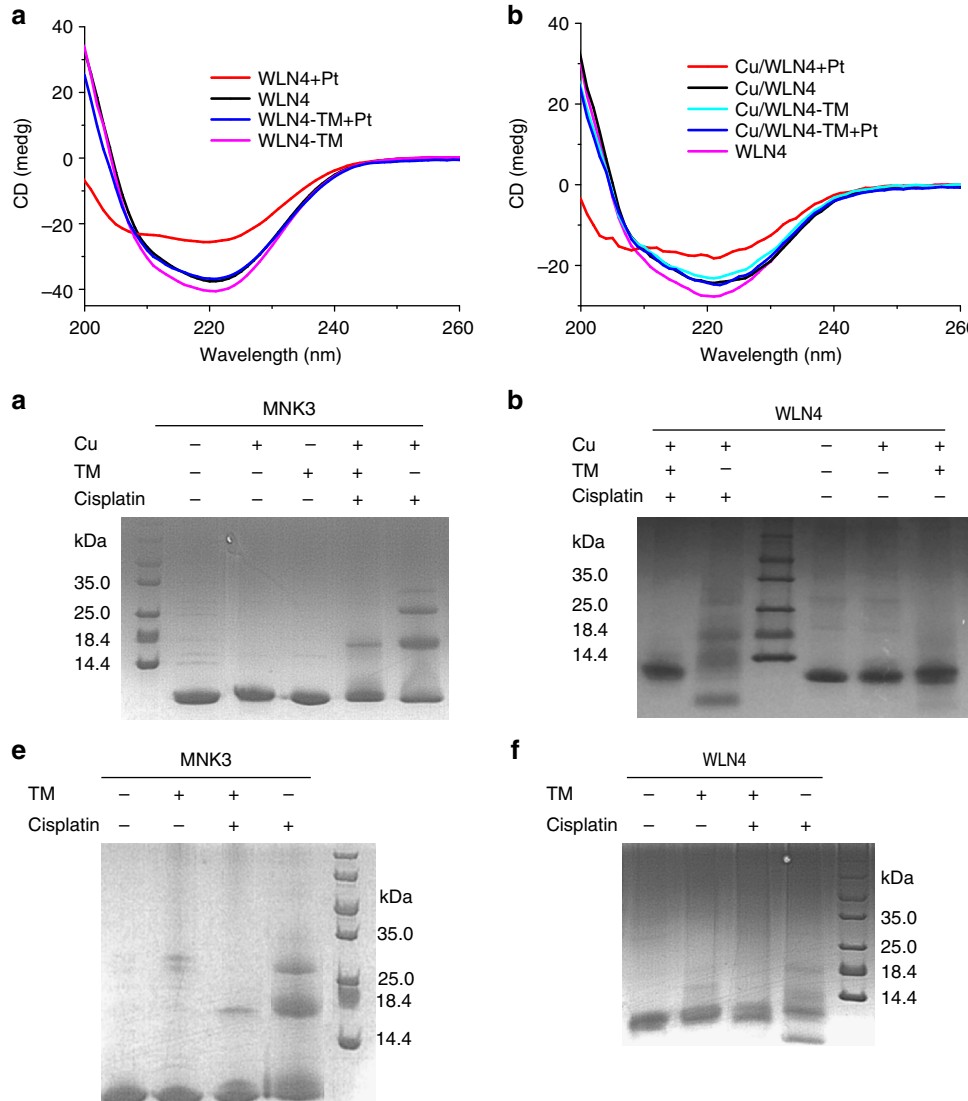

**Fig. 4** TM inhibition of the protein aggregation induced by cisplatin. **a**, **b** CD spectra of WLN4 (**a**) or Cu-WLN4 (**b**). TM-WLN4 complexes were prepared by reaction of the protein with 3 eq. TM for 3 h at 37 °C. The protein platination was performed by incubation with 3-molar equivalents of cisplatin for 8 h. **c–f** Gel electrophoresis analysis of the protein aggregation inhibited by TM. **c** Cu-MNK3; **d** Cu-WLN4; **e** apo-MNK3; **f** apo-WLN4. The copper protein was prepared by incubation with 1.5-fold Cu for 15 min at room temperature. Proteins (200 μM) were reacted with TM (600 μM) for 3 h, then were incubated with cisplatin (600 μM) for 2 h. Reactions were performed in 50 mM MES, 150 mM NaCl, pH 6.0 at 37 °C. Symbols +/- indicate presence/absence of the species

In addition, electron paramagnetic resonance (EPR) spectroscopic measurements were conducted to verify the effect of reducing conditions on the reaction of TM. Neither $(NH_4)_2MoS_4$ nor the final adduct of Mo-WLN4 gave an EPR signal, indicating that these Mo atoms are in an electron spin inactive Mo(VI) state. EPR spectra were also recorded during the reaction. Shortly after mixing $(NH_4)_2MoS_4$ with WLN4 (~0.5 min), an EPR signal corresponding to Mo(V) was observed in these spectra (Supplementary Fig. 11). Although the signal was weak, the g value (1.976) of the peak is consistent with the typical Mo(V) species[24,25]. The Mo(V) signal was generated in the reaction of WLN4 with TM in the presence of dithiothreitol (DTT), as well as glutathione or ascorbic acid, the two reducing agents abundant in cells. These results suggest that formation of the $Mo_2S_6O_2$ cluster observed herein could be due to differences in the redox environment from the formation of Atx1 adducts. The inclusion of 4.5 mM glutathione in the experiments is consistent with its concentration in cells (1−10 mM)[26]. In addition, this Mo-

centered copper-free adduct could be more relevant in the case of WLN since ATPases in the cytoplasm exist in apo-form, and copper-bound ATPases are localized at the cell membrane[27].

In summary, tetrathiomolybdate (TM) can react with the metal-binding domain of ATPases (WLN4) and form a $Mo_2S_6O_2$-centered protein dimer. Mo atoms directly bind to the thiol groups of the copper-binding residues of WLN4, and copper ions bound to WLN4 are expelled from the protein following Mo-coordination. The reaction of apo-WLN4 with TM generated an identical structure to that of Cu-WLN4. These results indicate that formation of the copper-free $Mo_2S_6O_2$ core plays a pivotal role in the formation of the dimeric TM-WLN4 adduct, which is rather different from the reaction of the copper chaperone Atx1 that generates a copper-linked TM-Cu-Atx1 ternary adduct. The reducing conditions in the TM reaction could be associated with the formation of the $Mo_2S_6O_2$ cluster that crosslinks two WLN4 molecules in the dimeric TM-WLN4 adduct. Interestingly, in the reaction of multi-domain proteins, TM can connect two

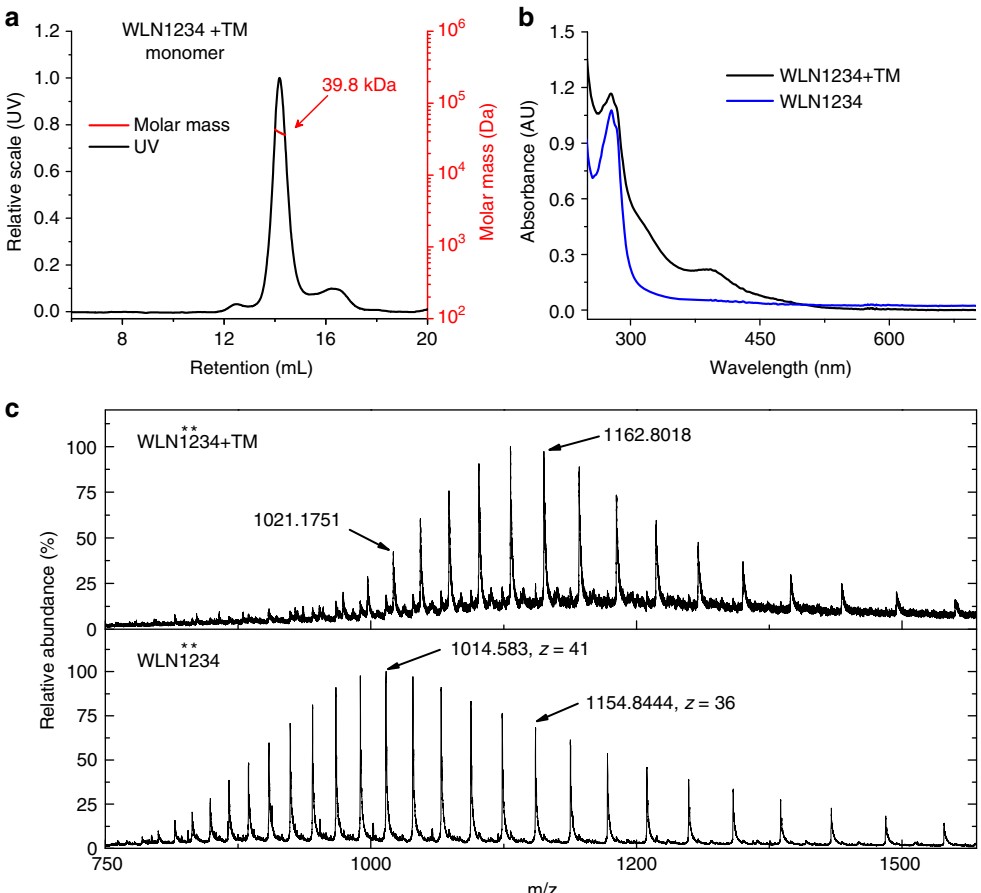

**Fig. 5** TM induces intramolecular crosslink of MBDs in WLN. **a** SEC-MALS analysis of the monomeric fraction obtained from the reaction of WLN1234 with TM. **b** UV–Vis measurement of WLN1234 and the monomeric adduct from the reaction with TM. **c** ESI-MS spectra of WLN1234 before (low portion) and after (up portion) the reaction with TM. A protein mutant of WLN1234 (Cys-to-Ser mutation on domains 1 and 2, MW 41541.07 Da) was used in ESI-MS analysis. Multiple peaks are present due to differently charged signals. The *m/z* of two representative peaks (+41 and +36 charged) are labeled in the figure

MBDs in the same protein via intramolecular crosslinking, which could lead to efficient inhibition of the functions of WLN protein. In addition, the formation of the TM-WLN4 dimer inhibits the reaction of ATPases with cisplatin. Since ATPases are associated with the sequestration of cisplatin, this finding reveals the molecular basis of TM in the sensitization of cellular responses to cisplatin. These results provide the structural and chemical insight into the mechanism of TM on copper proteins, and explain the synergy of TM and cisplatin in cancer chemotherapy.

## Methods

**Cloning and protein expression**. Genes of MBDs of ATPases were amplified from a human cDNA library by PCR, then inserted into expression vectors, including pST-SG for WLN4 and MNK3, pET-22b (+) for WLN456 and pET28a (+) for WLN1234 and all mutants of WLN1234 and WLN456. Plasmids were transformed into competent *Escherichia coli* BL21 (DE3) cells (Novagen), and cells were cultured at 37 °C with shaking overnight in Luria-Bertani medium containing 100 μg/mL ampicillin. Protein expression was induced with 0.4 mM isopropyl β-d-1-thiogalactopyranoside when the absorbance at 600 nm ($OD_{600}$) reached 0.6–0.8, and culturing was continued for 5 h at 37 °C. Cells were harvested by centrifugation, and proteins were purified by chromatography (see Supplementary Methods for details).

**Crystallization and X-ray diffraction data collection**. Crystals were grown at 13 °C using the sitting-drop vapor-diffusion method, with an initial mixing condition of 1 μL protein solution (2 mg/mL) and an equal volume of reservoir solution. Crystals of TM-WLN4 were obtained from 0.1 M MES and 0.2 M Ca (OAc)₂. All crystals were soaked in cryoprotectant buffer containing 25% glycerol (v/v) and flash-cooled in liquid nitrogen. X-ray diffraction data were collected at

synchrotron radiation beamline BL19U1, Shanghai Synchrotron Radiation Facility, using an ADSC QUANTUM 315R CCD detector with a crystal-to-detector distance of 220 mm for TM-WLN4 from Cu-WLN4, and 300 mm for TM-WLN4 from apo-WLN4. Individual frames were collected at 100 K using 1 s for each 1.0° oscillation over a range of 360° for both datasets. Diffraction data were indexed, integrated, scaled, and merged using the program HKL2000[28]. The details about structure determination and refinement were described in Supplementary Methods.

**Size-exclusion chromatography coupled with multi-angle light scattering**. SEC-MALS measurements were performed using an in-line DAWN HELEOS-II MALS detector (Wyatt Technology) and an Optilab T-rEX differential refractometer (Wyatt Technology) at room temperature. Samples were separated on a Superdex 75 10/300 GL column (GE Healthcare) in SEC-MALS buffer (50 mM MES pH 6.0, 150 mM NaCl) at a flow rate of 0.4 mL/min. The SEC-MALS data were analyzed by ASTRA 6.1 software.

**Protein platination analyses**. Cisplatin was prepared in 3.3 mM aqueous solution. Reactions of WLN4 with cisplatin were analyzed by NMR spectroscopy. NMR spectra were recorded on an Avance 600 MHz spectrometer (Bruker) at 298 K. ¹⁵N-labeled proteins (0.5 mM) were prepared in 50 mM sodium phosphate buffer (pH 7.4) containing 5% (v/v) D₂O. WLN4 was treated with 1.5 mM TM at 37 °C for 3 h. Then 1.5 mM cisplatin was added to the complex and incubated at 37 °C for 2 h. Data were processed and analyzed using TopSpin software.

Reactions of WLN4 with cisplatin were analyzed by electrospray ionization mass spectrometry (ESI-MS). Proteins were desalted using a desalting column and ESI-MS measurements were performed on a Thermo LTQ linear Orbitrap XL mass spectrometer (Thermo Fisher, San Jose, CA, USA). Data were recorded in an *m/z* range of 200 to 2000 in positive ion mode. Spectra were analyzed using Xcalibur 2.0 software (Thermo). Other detailed methods for platination analyses were described in Supplementary Methods.

## Data availability

The structures of TM-WLN4 from Cu-WLN and from apo-WLN4 have been deposited in the Protein Data Bank (PDB 6A71 and 6A72, respectively). A reporting summary for this Article is available as a Supplementary Information file. All data supporting the findings of this study are available within this article and its supplementary information files or from the corresponding author upon reasonable request.

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

## Acknowledgements

This work was supported by National Key R&D Program of China (2017YFA0505400, 2017YFA0503600, 2016YFA0400903) and the National Science Foundation of China (21573213, 21877103, U1532109, 31700671), the Strategic Priority Research Program of the Chinese Academy of Sciences (XDB19000000), the Foundation for Innovative Research Groups of the National Natural Science Foundation of China (31621002), and the Major Program of Development Foundation of Hefei Center for Physical Science and Technology (2018ZYFX004, 2017FXCX004). We gratefully acknowledge Shanghai Synchrotron Radiation Facility (SSRF), Core Facility Center for Life Science, USTC and Instruments' Center for Physical Science, USTC. A portion of this work was performed on the Steady High Magnetic Field Facilities, High Magnetic Field Laboratory, CAS.

## Author contributions

T.F. and W.C. coordinated the research and designed the experiments with support from Y.S. and S.Y. T.F. performed the experiments of protein reactions and analyzed the data with support from G.H. and J.S. W.C. acquired the X-ray of protein and processed data with the supervision of X.Z. and J.Z. Q.T. performed the EPR analysis and G.L. collected ESI-MS data. T.F. and W.C. wrote the first draft of the manuscript. Y.L. conceived and supervised the study and wrote the final version of the manuscript. All authors contributed to analysis and discussion of the data, and have reviewed and approved the manuscript.
