## [Peer Review File · Nature Communications]

Reviewers' comments:

Reviewer #1 (Remarks to the Author):

This is an interesting paper, and appropriate for publication after some modifications. The authors have found that tetrathiomolybdate (TM) can act as a bridge between forming a dimer of the metal binding domain in WLN4. While it is not necessarily all that surprising that a thiophilic metal such as Mo can bind to multiple thiolates from different monomers, forming a dimer, there are several novelties to this structure that may justify publication in Nature. One very interesting feature is that the behavior seen here is quite different from that seen for ATOX1, which has a broadly similar Cu site. In ATOX1, Cu-protein binding is retained and a single Mo serves as the bridge. In the present case, the Cu is lost and the Mo forms a dimer.

The authors note (page 1, bottom) that there is “no copper diffraction”. It would be better stated that there is no electron density peak that can be associated with copper – one doesn’t expect “copper diffraction” per se. However, I have bigger concern on this point. The absence of electron density that can be attributed to Cu does *not* mean that copper is not present – it simply means that the copper, if present, is not ordered (or, alternatively, that Cu is released from only a small fraction of protein, but it is that small fraction that is crystallized). The data in table 1 are far better evidence that Cu is released from most of the Cu-WLN4 on treatment with TM; the authors should rely on this and not the absence of identifiable Cu peaks in the diffraction.

I have two substantive concerns with this manuscript which should be addressed in a revised manuscript.

First, the authors speculate (page 5, top) that TM connect two metal-binding domains (MDB) in the same protein. I don’t see any evidence for this. I agree that it could happen, but demonstrating that two MDB fragments are able to bind to TM falls far short from showing that TM forms a bridge between two MDBs. First, the present experiments appear to all be using the MDB fragment WLN4 and not the intact ATP7B. It’s plausible that WLN4 (and the other MDBs) bind TM when they are part of ATP7B, but this doesn’t seem to have been shown. More importantly, there seems to be no evidence that the relative geometry of two different binding domains in ATP7B is such that the dimer can form. If there is such evidence, it should be included; otherwise, I think that this speculation should be omitted (or at least limited).

Secondly, I do not find the evidence for a Mo(V) intermediate to be compelling. No gains are stated, but based on the noise (e.g., Fig. S6) it appears that the Mo(V) signal must be very weak in the “GSH” and “Vc” samples compared to the DTT treated sample. The same holds for all 3 protein samples Figure 5. Without some information on spin quantitation, I suspect that this reflects the presence of a small minority of a species with unpaired spin (possibly Mo(V), although the agreement with the authentic DTT-treated sample is not great). Moreover, I don’t see any particular reason to assume that a Mo(V) intermediate would be on the pathway that converts the Mo(VI) in TM to the presumptive Mo(VI) in the product. Why not simple ligand exchange without any redox?

Minor comments

The manuscript is generally clear, but could benefit from the attentions of a copy editor. I believe that the acronym is MALDI-MS, not MOLDI-MS

I think that the authors mean 2Fo-Fc, not Fo-Fc (page 2).

The naming scheme that the authors use is rather confusing. Why is this first dimer called TM-Cu-WLN4 and the second called TM-WLN4. The available evidence suggests that there is no Cu

present, making the former name a bit odd. It's true that the authors began with a Cu-bound protein, but they report that Cu is lost. In fact, I see no reason to assume that their two structures are anything other than slightly different quality crystals of the same complex. If the authors believe that they are different, they should explain why. Otherwise, it might be appropriate to restructure the paper to simply indicate that the same crystalline species is obtained regardless of whether Cu is present initially.

It appears to me that the second domain in "TM-WLN4" is colored grey, not wheat (Fig. S3). If so, the caption and color boxes should both be changed.

Reviewer #2 (Remarks to the Author):

This is a well done study in which TM binding to a metal binding domain (WND4) of ATP7B has been studied by crystallography and solution methods. In contrast to TM binding with Atox1, which resulted in a trimer with Cu, here a dimer without Cu is found. The experiments are well done so the data appears solid.

I have some experimental comments for the results followed by comments on the conclusions. upon successfully amending these issues, i would recommend publication.

First, protein expression and purification needs better description. several different domains were prepared, including one from MNK, and these must be specified. are they all purified the same? what sequences did you include in the various constructs? how do each of these domains run on SEC?

also, it is not clear, why a MNK domain was used here.

how was proteins loaded with Cu(I)? and how was platinum solution prepared? this is crucial and can affect experiments a lot. but nothing is mentioned about this.

as the authors use NMR to characterize the protein and metal binding in solution, they should also use NMR to confirm dimer formation in solution. SEC of these small proteins is not particularly accurate. for example, although Atox1 and WND domains have the same size, WND4 runs as a larger protein than Atox1. thus i am not convinced that the solution state is a dimer with TM.

for the conclusion part, the authors must think again.

in the current text they propose that the 'different structures and functions' of atox1 (small, soluble) and atp7b (multi-domain, membrane bound) explain why one (atox1) interacts as a trimer with cu with TM and the other (WND4) forms a dimer without Cu with TM. This is not a correct statement as all the experiments were made with single domains of WND4. thus, from a structural point of view, in the experiments, atox1 and wnd4 should behave the same. as mentioned in the text, atox1 and wnd4 have the same fold and Cu site. the authors did not do any experiments with full length atp7b. so there must be another reason for the discrepancy in their two different studies.

one may wonder if both proteins could form both types of complexes and it is just a matter of getting a range of different crystals? this is the key point, to find a reasonable explanation for the results!!

another point is the unspoken conclusion one can make from this study that instead of chelating Cu, TM binds to proteins and makes them expel Cu, thus creating additional free Cu (opposite of a chelator). This topic should be discussed.

In the experiments here, could one argue that TM could

bind WND4 and expel Cu and then another TM chelates that released Cu? thus TM could have dual functions? did any experiments address this? can concentration dependencies say something about this?

finally, the English language and grammar needs improvement.

Reviewer #1:

This is an interesting paper, and appropriate for publication after some modifications. The authors have found that tetrathiomolybdate (TM) can act as a bridge between forming a dimer of the metal binding domain in WLN4. While it is not necessarily all that surprising that a thiophilic metal such as Mo can bind to multiple thiolates from different monomers, forming a dimer, there are several novelties to this structure that may justify publication in Nature. One very interesting feature is that the behavior seen here is quite different from that seen for ATOX1, which has a broadly similar Cu site. In ATOX1, Cu- protein binding is retained and a single Mo serves as the bridge. In the present case, the Cu is lost and the Mo forms a dimer.

Response:

We thank the Reviewer's comments *that there are several novelties to this structure that may justify publication in Nature*. We have revised the manuscript according to the Reviewer's suggestions.

The authors note (page 1, bottom) that there is "no copper diffraction". It would be better stated that there is no electron density peak that can be associated with copper – one doesn't expect "copper diffraction" per se. However, I have bigger concern on this point. The absence of electron density that can be attributed to Cu does not mean that copper is not present – it simply means that the copper, if present, is not ordered (or, alternatively, that Cu is released from only a small fraction of protein, but it is that small fraction that is crystallized). The data in table 1 are far better evidence that Cu is released from most of the Cu-WLN4 on treatment with TM; the authors should rely on this and not the absence of identifiable Cu peaks in the diffraction.

Response:

We thank the Reviewer's suggestion. We revised the manuscript by using Table 1 as the proof of copper ejection from the TM-WLN4 product. In order to further confirm the absence of copper in the TM-WLN4 crystal, we measured the amount of Cu and Mo in the crystal of TM-WLN4, which was prepared with Cu-WLN4. By dissolving crystals, the contents of Cu and Mo in the solution were analyzed using ICP-MS. The result clearly showed that, while Mo is present in the crystal, no Cu was detected (see following Table R1). This result supports the conclusion that TM binding expel Cu from the protein. These data were added in the revised manuscript (page 1, the last paragraph on the right column).

Table R1. Cu and Mo measured in the crystal of TM-WLN4*.

Cu	0.000 μ M
Mo	0.039 μ M

* The crystal was obtained from the reaction of TM with Cu-WLN4.

In addition, we also collected the single-wavelength anomalous diffraction (SAD) of the TM-WLN4

crystal at 1.3808 Å, which is the wavelength of Cu-K edge energy. The anomalous signal of Cu was determined by using Xtriage in Phenix. The anomalous signal of Cu is quite weak, and we failed to find the location of any Cu atoms by using shelx C/D in CCP4i, further supporting that Cu is not present in the crystal (Figure R1).

Figure R1. The anomalous signal of Cu was determined by using Xtriage in Phenix.

I have two substantive concerns with this manuscript which should be addressed in a revised manuscript.

First, the authors speculate (page 5, top) that TM connect two metal-binding domains (MDB) in the same protein. I don't see any evidence for this. I agree that it could happen, but demonstrating that two MDB fragments are able to bind to TM falls far short from showing that TM forms a bridge between two MDBs. First, the present experiments appear to all be using the MDB fragment WLN4 and not the intact ATP7B. It's plausible that WLN4 (and the other MDBs) bind TM when they are part of ATP7B, but this doesn't seem to have been shown. More importantly, there seems to be no evidence that the relative geometry of two different binding domains in ATP7B is such that the dimer can form. If there is such evidence, it should be included; otherwise, I think that this speculation should be omitted (or at least limited).

Response:

Thanks for the comments. In order to address the Reviewer's question, we constructed multi-domain proteins of WLN, including the domains 1-4 (WLN1234) and domains 4-6 (WLN456). SEC analyses showed that, upon TM reaction, WLN1234 forms both monomeric and oligomeric adducts. It is not surprising that the oligomeric species were generated due to the intermolecular crosslinking of these multi-domain proteins; however, the monomeric species could contain Mo-connected intramolecular crosslinking. Therefore, the fraction appeared at the retention of monomer species was collected for further analyses (see following the Figure R2A). SEC-MALS analysis indicated that this fractions exhibited molecular mass of 39.8 KDa, confirming the monomeric species of this adduct TM-WLN1234 (cal. 40.4 KDa) (Figure R2B). The presence of Mo-coordination was confirmed by UV-vis spectra, as the monomeric TM-adduct exhibits absorption at ~390 and ~320 nm due to the Mo-

coordination, which is consistent with the TM-WLN4 complex (Figure R2C and R2D for WLN1234 and WLN4, respectively). Moreover, mass spectra provided direct evidence that the TM reaction led to the mass increase by 287.4 Da on WLN1234, which is well in agreement with the addition of $\text{Mo}_2\text{S}_2\text{O}_2$ (cal. 288.0 Da) to the protein (Figure R2E). These data confirmed that the binding of $\text{Mo}_2\text{S}_6\text{O}_2$ cluster with in the WLN1234 molecule. In addition, the TM reaction was also conducted on WLN456. The SEC-MALS and UV-vis results also showed the monomeric species of WLN456 with TM binding. These results indicate that the formation of intramolecular $\text{Mo}_2\text{S}_6\text{O}_2$ cluster between two MBDs in the WLN1234 and WLN456 proteins.

To further locate the two Mo-crosslinked domains in the multi-domain proteins, we performed mutagenesis of different MBDs (Figure S2), as the Cys-to-Ser mutation abolishes the Mo coordination to the protein domain. By comparing the UV-vis spectra of different variants, we can conclude that the domain 3 and domain 4 are the two Mo-linked domains in WLN1234, as the mutation of domains 1 and 2 does not influence the Mo binding, whereas no Mo-absorption was observed if either domain 3 or 4 was mutated (Figure R2F). For the same reason, the Mo-induced crosslink occurs between the domain 5 and domain 6 in WLN456 (Figure R2G). In addition, it has been postulated that, while transporting copper ions, WLN4 could deliver Cu(I) to WLN3 and WLN6 through copper linked interdomain connections (*PNAS*, 2006,103(15) 5729–5734; *Biochemistry* 2008, 47(28), 7423–7429), suggesting that the long flexible linker between MBDs allows the geometry for interdomain interaction.

These new results were added in the last two paragraphs of the Result Section of the revised manuscript. (see Page 4, the two paragraphs above Fig. 5)

Figure R2. TM induces intramolecular crosslink of MBDs in WLN. (A) SEC profiles of the products from the reaction of TM with WLN1234 or its mutants. (B) SEC-MALS analysis of the monomeric fraction obtained from the reaction of WLN1234 with TM. (C) UV-vis measurement of WLN1234 and its monomeric adduct from the reaction with TM. (D) UV-vis spectra of WLN4 and the TM-WLN4 adduct. (E) ESI-MS spectra of WLN1*2*34 before (low portion) and after (up portion) the reaction with TM. Multiple peaks are present due to differently charged signals. The m/z (+41 and +36) of two representative peaks are labeled in the figure. (F-G) UV-vis spectra of the TM adducts of multi-domain proteins (F: WLN1234, G: WLN456). All adducts from multi-domain proteins were purified by SEC, and the fractions of monomeric species were collected for UV-vis analysis. Asterisks denote the mutated domains in which the metal binding site CXXC was replaced by SXXS.

Secondly, I do not find the evidence for a Mo(V) intermediate to be compelling. No gains are stated, but based on the noise (e.g., Fig. S6) it appears that the Mo(V) signal must be very weak in the “GSH” and “Vc” samples compared to the DTT treated sample. The same holds for all 3 protein samples Figure 5. Without some information on spin quantitation, I suspect that this reflects the presence of a small minority of a species with unpaired spin (possibly Mo(V), although the agreement with the authentic DTT-treated sample is not great). Moreover, I don’t see any particular reason to assume that a Mo(V) intermediate would be on the pathway that converts the Mo(VI) in TM to the presumptive Mo(VI) in the product. Why not simple ligand exchange without any redox?

Response:

Thanks for the suggestion. We agree with the Reviewer that the evidence of Mo(V) intermediate is not compelling, and it is not necessary to assume that a Mo(V) intermediate is on the pathway.

According to the Reviewer's suggestion, we removed the statement and discussion about the Mo(V) intermediate, and moved the figure to Supporting Information only to show that the EPR signal was detected during the reaction. This EPR result was cited in the Discussion Section to compare the reaction condition to the formation of trimeric TM-Atx1 adducts. (Page 5, the last paragraph of Discussion on the left column)

Minor comments

- The manuscript is generally clear, but could benefit from the attentions of a copy editor.

Response:

We thank the Reviewer for the detailed suggestions. We have checked the whole manuscript and corrected grammar and typo errors.

- I believe that the acronym is MALDI-MS, not MOLDI-MS

Response:

The term of MALDI-MS was corrected.

- I think that the authors mean 2Fo-Fc, not Fo-Fc (page 2).

Response:

For the electron density map, we actually deleted the ligand and generated the *Fo-Fc* omit map but not *2Fo - Fc* map.

- The naming scheme that the authors use is rather confusing. Why is this first dimer called TM-Cu-WLN4 and the second called TM-WLN4. The available evidence suggests that there is no Cu present, making the former name a bit odd. It's true that the authors began with a Cu-bound protein, but they report that Cu is lost. In fact, I see no reason to assume that their two structures are anything other than slightly different quality crystals of the same complex. If the authors believe that they are different, they should explain why. Otherwise, it might be appropriate to restructure the paper to simply indicate that the same crystalline species is obtained regardless of whether Cu is present initially.

Response:

We thank the Reviewer for the suggestions. The name of the protein structure was changed to TM-WLN4. In the case of comparison two structures from Cu-WLN4 and apo-WLN4, we used the name TM-WLN4^a and TM-WLN4^b to distinguish the two different structures, and gave annotations for clarify the protein complexes.

- It appears to me that the second domain in "TM-WLN4" is colored grey, not wheat (Fig. S3). If so, the caption and color boxes should both be changed.

Response:

The color of the second domain in TM-WLN4 has been changed to gray, and color annotation was amended in the revised manuscript.

Reviewer #2 (Remarks to the Author):

This is a well done study in which TM binding to a metal binding domain (WND4) of ATP7B has been studied by crystallography and solution methods. In contrast to TM binding with Atox1, which resulted in a trimer with Cu, here a dimer without Cu is found. The experiments are well done so the data appears solid.

Response:

We thank the Reviewer's comments *that 'The experiments are well done so the data appears solid'*. We have revised the manuscript according to the Reviewer's suggestion.

I have some experimental comments for the results followed by comments on the conclusions. upon successfully amending these issues, i would recommend publication.

First, protein expression and purification needs better description. several different domains were prepared, including one from MNK, and these must be specified. are they all purified the same? what sequences did you include in the various constructs? how do each of these domains run on SEC?

also, it is not clear, why a MNK domain was used here.

Response:

The expression and purification of each protein have been described in more detail in the revised manuscript, including SEC profiles of the purification. The protein sequences of each domain have been provided.

A MNK domain (MNK3) was used since both WLN and MNK proteins are associated with the drug resistance of cisplatin. Therefore, we have investigated the effect of TM on the platination of the metal binding domain of WLN and MNK. We have revised the statement in the revised manuscript to explain this reason. (Page 4, the first paragraph after Fig. 4 on the left column)

how was proteins loaded with Cu(I)? and how was platinum solution prepared? this is crucial and can affect experiments a lot. but nothing is mentioned about this.

Response:

The Cu(I) was loaded by incubation of proteins with $[\text{Cu}(\text{CH}_3\text{CN})_4]\text{ClO}_4$. The solution of cisplatin was prepared by dissolving of cisplatin in water at a final concentration of 3.3 mM. The following

experimental details were added in the revised manuscript.

Supporting Information: *The Cu-WLN4 was prepared by the incubation of the purified WLN4 protein with 1.5-molar ratio of [Cu(CH₃CN)₄]ClO₄ for 15 min at room temperature.*

Methods Section: *Cisplatin was prepared in 3.3 mM aqueous solution.*

as the authors use NMR to characterize the protein and metal binding in solution, they should also use NMR to confirm dimer formation in solution. SEC of these small proteins is not particularly accurate. for example, although Atox1 and WND domains have the same size, WND4 runs as a larger protein than Atox1. thus i am not convinced that the solution state is a dimer with TM.

Response:

We thank the Reviewer's comment on the SEC data and reminded "WND4 runs as a larger protein than Atox1". To avoid possible errors from SEC, the multi-angle light scattering measurement (MALS) was used in cooperation with SEC. The molecular weight obtained from SEC-MALS measurements indicated that the TM-WLN4 adduct is about 15.0 ~ 15.5 kDa, which is corresponding to the dimer of WLN4 (~ 7.5 kDa from the MALS measurement). Additionally, the MALDI-TOF mass spectra confirmed the formation of dimeric TM-WLN4 product in the reaction of apo-WLN4 (m/z: cal. 15607 Da, obs. 15588 Da, see Figure S4A).

According to the Reviewer's suggestion, NMR relaxation measurements were performed to analyze the polymerization status of WLN4 upon TM binding. The rotational correlation time (τ_c) of a protein in solution is related to the longitudinal (T_1) and transverse (T_2) ¹⁵N relaxation times and nuclear frequency ($\nu_N = 60$ MHz) according to Eq. 1 (*J Biomol. NMR*, 2010, 46, 11-22)

$$\tau_c \approx \sqrt{\left(\frac{6T_1}{T_2} - 7\right)} / 4\pi\nu_N \quad (1)$$

The average values of T_1 and T_2 of proteins were obtained by fitting the integrated signals of the backbone amide on the HSQC spectra, which gave two different vales of τ_c (see following Table R2). The molecular weight was estimated from τ_c according to their correlation (*J Biomol. NMR*, 2010, 46, 11-22).

Table R2. Estimation of the molecular size of the TM-WLN4 complex according to NMR relaxation

	T_1 (ms)	T_2 (ms)	τ_c (ns)	MW (kDa)
WLN4	453 ± 40	142 ± 19	4.62	7.7
TM-WLN4	830 ± 124	72 ± 20	10.44	17.4

These data were added in the revised manuscript (page 2, the first paragraph on the right column).

for the conclusion part, the authors must think again.

in the current text they propose that the 'different structures and functions' of atox1 (small, soluble) and atp7b (multi-domain, membrane bound) explain why one (atox1) interacts as a trimer with Cu with TM and the other (WND4) forms a dimer without Cu with TM. This is not a correct statement as all the experiments were made with single domains of WND4. thus, from a structural point of view, in the experiments, atox1 and wnd4 should behave the same. as mentioned in the text, atox1 and wnd4 have the same fold and Cu site. the authors did not do any experiments with full length atp7b. so there must be another reason for the discrepancy in their two different studies.

one may wonder if both proteins could form both type of complexes and it is just a matter of getting a range of different crystals? this is the key point, to find a reasonable explanation for the results!!

Response:

We thank the Reviewer for the suggestion about different structure of WLN4 and Atox1 adducts. In order to address this question, we have conducted the reaction of TM with two proteins in the identical condition in this work. The SEC-MALS analysis indicates that both apo-Atox1 and Cu-Atox1 generated dimeric adducts in the reactions of TM (Figure S10). This result showed that TM can induce the dimerization of both WLN4 and Atox1, and copper is not required in the reaction. Therefore, the different structure of two protein adducts (the Cu-Mo-centered trimeric TM-Atox1 in the literature and the copper-free dimeric TM-WLN4 in this work) could be due to different redox environments of two reactions. TM-Atox1 was prepared in aerobic condition in the literature, while TM-WLN4 was generated in the presence of reducing agent in this work. It is worth noting that intracellular environment is in reducing condition. The concentration of reducing agent (4.5 mM) used in this work is in well agreement with the concentration of cellular glutathione (1 - 10 mM). Therefore, the copper-free dimeric adducts could be more relevant to the physiological conditions.

This result was discussed in the revised manuscript in the second paragraph of the Discussion section.

another point is the unspoken conclusion one can make from this study that instead of chelating Cu, TM binds to proteins and make them expel Cu, thus creating additional free Cu (opposite of a chelator). This topic should be discussed.

In the experiments here, could one argue that TM could bind WND4 and expel Cu and then another TM chelates that released Cu? thus TM could have dual functions? did any experiments address this? can concentration dependencies say something about this?

Response:

To verify whether additional TM could chelate with the copper ions released from WLN4, we measured UV spectra of the proteins after TM binding. In comparison to the reaction of apo-WLN4,

the reaction of Cu-WLN4 with TM resulted in additional shoulder peaks at ~ 400 nm and 520 nm (Figure R3A). This difference is consistent with the reaction of TM with free Cu ions, which also generates a shoulder peak at this region relative to free TM (Figure R3B). This result suggests that, while TM binds to WLN4 and expels copper ions from the proteins, additional TM can chelate the copper ions released from the protein. This Cu-TM chelation could enhance the protein reaction by promoting copper release. This result is consistent with the ICP-MS measurement that equimolar of TM can cause only 45% copper release from Cu-WLN4 (Table 2), as a portion of TM chelates with the copper ions released from the protein. The data indicate that TM has a dual-function in the reaction with Cu-WLN4. This point has been discussed in the revised manuscript (page 3, the paragraph above Table 2 on the left column). We thank the Reviewer's suggestion.

Figure R3. UV-vis spectra of TM in the reaction with copper ions or WLN4. (A) The spectra recorded on reaction of TM with apo-WLN4 (red) or Cu-WLN4 (magenta). The reactions were performed on $60 \mu\text{M}$ TM with $30 \mu\text{M}$ protein at 37°C for 2 h. (B) The spectra of $60 \mu\text{M}$ TM in the absence (black) or in the presence of Cu(I) ions ($30 \mu\text{M}$ blue). It can be observed that, upon Cu(I) reaction, the characteristic peaks of TM at 317 nm and 468 nm decreased, while two shoulder peaks appeared at ~ 400 nm and 520 nm from the Cu-TM complex.

finally, the english language and grammar needs improvement.

Response:

Thanks for the suggestion, we have checked the whole manuscript and corrected grammar and typo errors. We hope the revised manuscript will be found suitable for publication in *Nature Communications*.

REVIEWERS' COMMENTS:

Reviewer #1 (Remarks to the Author):

The revised manuscript addresses the scientific concerns that I had. I find this a much improved paper and recommend publication.

I would note that the manuscript would benefit from the attention of a careful copy-editor. One example, of many -- I suspect that in the conclusions, the authors wish to refer to "pivotal role", not the "pivot rule" of TM in dimer formation.

Reviewer #2 (Remarks to the Author):

The revisions answer my questions. I recommend publication.

REVIEWERS' COMMENTS:

Reviewer #1 (Remarks to the Author):

The revised manuscript addresses the scientific concerns that I had. I find this a much improved paper and recommend publication.

I would note that the manuscript would benefit from the attention of a careful copy-editor. One example, of many -- I suspect that in the conclusions, the authors wish to refer to "pivotal role", not the "pivot rule" of TM in dimer formation.

Response:

Thanks for the suggestion. The manuscript has been polished by an English language editing company. We have also carefully checked the whole manuscript to correct grammar/typo errors. Please find corrections in the revised manuscript.

Reviewer #2 (Remarks to the Author):

The revisions answer my questions. I recommend publication.

Response:

Thanks for the suggestion.